# Phosphoenolpyruvate and Related Metabolic Pathways Contribute to the Regulation of Plant Growth and Development

**DOI:** 10.3390/ijms26010391

**Published:** 2025-01-04

**Authors:** Runzhou Hu, Haiyang Yu, Jing Deng, Shanjing Chen, Ronglan Yang, Hongjun Xie, Xiao Tang, Yaying Yu, Yonghong Duan, Meng Zhang, Mingdong Zhu, Yinghong Yu

**Affiliations:** 1Long Ping Branch, College of Biology, Hunan University, Changsha 410125, China; pamelaparrish10232@gmail.com (R.H.); yuhaiyang@hnu.edu.cn (H.Y.); zhangmeng2019@hnu.edu.cn (M.Z.); 2State Key Laboratory of Hybrid Rice, Hunan Rice Research Institute, Hunan Academy of Agricultural Sciences, Changsha 410125, China; dengjing@hunaas.cn (J.D.); shanjing@hun.edu.cn (S.C.); yangronglan@hnu.edu.cn (R.Y.); xhj1110@126.com (H.X.); talent_tdx@hotmail.com (X.T.); yuyaying@hunaas.cm (Y.Y.); zzk126@126.com (Y.D.)

**Keywords:** phosphoenolpyruvate, carbon allocation, energy metabolism, plant growth and development

## Abstract

Phosphoenolpyruvate (PEP) plays a key role in the development of plants and exists in a wide variety of species. Research on the metabolic activities of PEP in plants has received increasing attention. PEP regulates multiple processes in plant growth and development. This article provides a comprehensive summary of these pathways, including embryo formation, root development, synthesis of secondary metabolites, and the formation of lignification. We also summarize new findings, including PEP’s role in nodule energy sensing and carbon allocation under the influence of ozone. This review displays the complex and differential regulatory pathways in plant growth and development and provides a reference for basic and applied research on PEP metabolism in plants.

## 1. Background

Energy metabolism is the basis of life activities and affects the growth and development of organisms. Carbon skeletons must be synthesized for organisms to complete normal biochemical metabolism. Phosphoenolpyruvate (PEP), which is an important intermediate product during the synthesis of carbon skeletons and energy metabolism, plays a key role in the growth and development of most organisms. In bacteria, PEP acts as a source of phosphoryl groups during the conversion of glucose to 6-phosphoglucose and the phosphorylation of various sugars and their derivatives through a phosphoric acid cascade reaction prior to their transport into cells. It also provides energy for life activities. In plants, it is involved in energy metabolism [1,2], the biosynthesis of various aromatic compounds, and carbon fixation during photosynthesis, while also participating in responses to external stresses (e.g., ozone stress) and root nodule energy state perception activities [3,4]. PEP (C_3_H_4_O_6_P) has a relative molecular mass of 267.22. Notably, it contains the highest-energy phosphate bond in living organisms (−61.9 kJ/mol) and exists mainly in an anionic form in living organisms [5].

As Shown in Figure 1. In plants, the PEP synthesis pathway is regulated by enolase (ENO), pyruvate, phosphate dikinase (PPDK), and phosphoenolpyruvate carboxykinase (PEPCK/PCK). The PEP metabolic pathway is regulated by pyruvate kinase (PK), 3-Deoxy-D-arabinoheptulosonate 7-phosphate synthase (DHAPS), and phosphoenolpyruvate carboxylase (PEPC). It is also regulated by the PEP/phosphorus translocator (PPT). Interestingly, it affects the energy state of plants as a signaling molecule. The following sections provide an overview of the overall regulatory effects of PEP involving the above-mentioned regulatory mechanisms.

### 1.1. Effects of ENO-Mediated PEP Synthesis on Plant Growth and Development

PEP synthesis mediated by ENO is a crucial step during glycolysis, which represents the initial sugar metabolism stage in all living cells. ENO serves as the rate-limiting enzyme that catalyzes the conversion of 2-PGA to PEP in the glycolytic pathway. Accordingly, ENO is essential for normal plant growth and development as well as for plant biochemical metabolism. In *Arabidopsis thaliana* (Arabidopsis), ENO exists as three distinct isoenzymes (ENO1, ENO2, and ENO3), of which ENO1 is targeted to the plastid in most heterotrophic tissues, but not in photosynthetic tissues. In a previous study, two T-DNA insertion *eno1* mutants had distorted trichomes and fewer root hairs than the wild-type (WT) control, but the mutation did not alter plant development and metabolism at the macroscopic level [7]. In Arabidopsis, the *At2g36530* (*ENO2*, also referred to as *LOS2*) gene encodes the ENO2 protein. Mutating this gene results in mutant plants with various severe cellular and growth defects, including small leaves and short siliques (i.e., dwarfism) and increased sensitivity to abiotic stresses. Additionally, these plants accumulate more salicylic acid (SA) than WT plants [8,9,10]. Lignin synthesis depends on PEP, especially during late developmental stages. The reported decrease in lignin contents suggests mutating *LOS2*/*ENO2* affects PEP synthesis. Moreover, decreases in SA levels may be due to the collapse of improperly lignified cell walls, resulting in the induction of an endogenous defense response, thereby promoting SA synthesis [11]. These reactions reflect the importance of a regular supply of PEP for the synthesis of secondary metabolites and normal plant development and metabolism. In addition to ENO, the *LOS2*/*ENO2* locus also encodes the transcription factor AtMBP-1 [12], which inhibits *LOS2*/*ENO2* promoter activity. This dual functionality helps to maintain an appropriate ENO2 activity level [13]. However, a new hypothesis posits that LOS2 does not participate in the immune response by replacing MBP-1. Rather, LOS2 decreases the classical ENO activity, which disrupts glycolysis and consequently activates the immune response involving NOD-like receptors. This, in turn, increases the accumulation of sugars and organic acids while depleting glycolytic metabolites, which ultimately affect plant growth. This is consistent with decreases in 3-phosphate, PEP, and relative total content of 2-phosphoglyceric acid and 3-phosphoglyceric acid levels in the *los2-2* mutant (relative to the corresponding contents in the WT control). Compared with the WT control, the *los2-2* mutant has shorter siliques, less chlorophyll, and flowers earlier, underscoring the intricate interconnections between primary metabolism and plant immunity [14]. In all organs, *ENO3* is consistently expressed at lower levels than *ENO1* or *ENO2*. In Arabidopsis, ENO3 is undetectable throughout the seed development stage. Furthermore, when expressed as a soluble full-length protein in an *Escherichia coli* ENO-deficient strain, ENO3 is inactive. However, if the ENO3 amino acid sequence is modified by replacing glutamic acid at position 251 with aspartic acid and aspartic acid at position 252 with glutamic acid, the ENO-deficient *E. coli* strain can grow normally. Thus, differences in the amino acid residues at the active site of ENO3 are the underlying cause of the observed lack of activity in *E. coli* [15].

### 1.2. Effects of PEPCK-Mediated PEP Synthesis and Degradation on Plant Growth and Development

Gluconeogenesis, which is mediated by PEPCK, occurs in most plants. Glycolysis synthesizes glucose from specific non-carbohydrate carbon substrates. This is a biochemical metabolic process common among diverse organisms, including fungi, yeast, bacteria, and other microorganisms. PEPCK plays a pivotal role in the glycolytic pathway. In the initial stage of glycolysis, when ATP is present, PEPCK is decarboxylated and phosphorylates oxaloacetic acid (OAA), thereby converting it to PEP. The transfer of phosphate results in the production of an ADP molecule. PEPCK directs the carbon released from fatty acid reserves to form sugars during seed germination until the photosynthetic apparatus is fully developed [16,17]. Numerous studies confirm that PEPCK affects plant growth and development, for example, disrupting *PEPCK* expression has detrimental effects on plant growth and development. Hypocotyl lengths in 5 d old 35S-*PCK1* antisense seedlings were reduced by up to 68% [18]. In transgenic tomato plants, the suppression of *PEPCK* expression via RNA interference inhibits root growth and decreases organic acid and soluble sugar contents. The decrease in the sugar content and accumulation of malate may be attributed to the inhibition of the gluconeogenesis pathway due to PEPCK deficiency [19]. In *Capsicum annuum*, *CaPEPCK*-overexpressing plants grow faster and produce larger leaves than WT plants. Moreover, CaPEPCK contributes to plant immune processes. Homozygous T-DNA insertion Arabidopsis mutants *pck1-3* and *pck1-4* are more susceptible to a pathogenic bacterium and a pathogenic oomycete than WT plants. Considered together, these results indicate that CaPEPCK enhances the innate immunity of plants against hemibiotrophic bacteria and obligate biotrophic oomycetes. Levels of glycolytic components in glycolysis activated by CaPEPCK1 overexpression may regulate defense signaling in plants [20]. For example, There is experimental evidence that levels of glycerol-3-phosphate are associated with basal defenses of the semi-living trophic fungus *Colletotrichum higginsianum* in *Arabidopsis thaliana* [21].

In plants, PEPCK is regulated by phosphorylation. Light significantly influences the phosphorylation of PEPCK. In *Zea mays* (maize) leaves, PEPCK is phosphorylated in dark-adapted leaves and dephosphorylated in light-adapted leaves. Phosphorylation and dephosphorylation decrease and increase PEPCK activity, respectively [22,23,24]. Sufficient illumination regulates the dephosphorylation of PEPCK. In maize, the extent of the phosphorylation at 11 sites of ZmPEPCK1 and at all sites of ZmPEPCK2 markedly increases as the light intensity increases, reaching considerable levels [25]. Additionally, PEPCK is phosphorylated in the cotyledons and endosperm of germinated seeds. Earlier research showed the dephosphorylation of PEPCK in *Cucumis sativus* (cucumber) cotyledons is stimulated by light [26]. Four phosphorylated residues (Ser-55, Thr-58, Thr-59, and Thr-120) were identified in maize ZmPEPCK1; the phosphorylation of these four residues is positively regulated by light. Moreover, mass spectrometry has been used to investigate post-translational phosphorylations. Both AthPEPCK1 and AthPEPCK2 are phosphorylated at multiple sites. A phosphomimic mutation at the Ser-62 site leads to increased enzymatic activity, whereas a mutation at the Thr-56 site results in decreased enzymatic activity. In contrast, a phosphomimetic mutation at the Thr-66 site does not alter enzymatic activity. This further supports the hypothesis that Athpepck1 is regulated by a complex mechanism involving a variety of signals, including those mediated by effectors and post-translational modifications [27].

Furthermore, the N-terminus of PEPCK is targeted for proteolysis in a process that is influenced by pH (i.e., decreased activity at pH 9–10). Additionally, protease inhibitors have no inhibitory effect on this process, and the quaternary structure of the enzyme remains unaltered [26,28,29]. In Arabidopsis, AthPEPCK1 is targeted by the cysteine protease METACASPASE9 (AthMC9) [30], which is present in the nucleus, cytoplasm, and peroxisome [30,31,32]. This enzyme is involved in the regulation of cell death in various physiological contexts, including immune responses [33,34] and vascular tissue development [35,36]. Specifically, the hydrolysis of the N-terminal domain of PEPCK appears to promote PEPCK activity. This is supported by a decrease in PEPCK activity in the crude extract of the Arabidopsis *mc9* mutant, as well as an increase in PEPCK activity in a 35S:*MC9*-overexpressing strain [30]. A comparison with the WT enzyme indicated that a salient characteristic of the truncated AthPEPCK1 is the altered allosteric regulation of metabolites. Glucose-6-phosphate (Glc6P) functions as an inhibitor of the WT enzyme but a weak activator of the Δ101 truncated form, whereas malate activates the WT enzyme but has no effect on the Δ101 truncated form [29]. Furthermore, both of these earlier studies suggest that the structural characteristics of the N-terminal are associated with the regulatory properties of PEPCK [37].

PEPCK is precisely regulated by many metabolites. In Arabidopsis, AthPEPCK1 and AthPEPCK2 are primarily inhibited by Glc6P, shikimate, and inorganic pyrophosphate (PPi) but activated by malate. The allosteric regulation of AthPEPCK1 and AthPEPCK2 by key metabolic intermediates, namely succinate, fumarate, citrate, and α-ketoglutarate, has been demonstrated. Intriguingly, malate and Glc6P activate and inhibit AthPEPCK1, respectively, but have no differential effect on AthPEPCK2 [38]. The regulatory effect of malate may be critical for stimulating the entry of carbon released via lipid degradation into the glycolytic pathway during seed germination. Once the photosynthetic apparatus is fully developed, increases in PEP and hexose phosphate levels may result in the inhibition of PEPCK, thereby limiting glycolysis. Furthermore, AthPEPCK1 is inhibited by shikimate, which is the precursor of aromatic amino acids and defense compounds. This regulation occurs during the synthesis of PEP, which is the initial substrate of the shikimate pathway [38]. The regulation of PEPCK by malic acid and Glc6P is the inverse of the regulation of PEPC. This may represent an important regulatory mechanism involving two enzymes in the plant cytoplasm that catalyze opposite reactions. This regulation may be the key to preventing an inefficient carboxylation/decarboxylation cycle, which depletes cytoplasmic ATP [38,39,40].

### 1.3. Effects of PPDK-Mediated PEP Synthesis and Degradation on Plant Growth and Development

PPDK plays a crucial role in the synthesis of PEP in plants. The plant C4 pathway represents one of three recognized mechanisms mediating carbon fixation during photosynthesis. PPDK catalyzes the conversion of pyruvate to PEP in the presence of ATP and phosphate. In plants, this reaction, which is reversible, occurs in the gluconeogenesis and C4 pathways [41]. PPDK is essential for regenerating PEP in C4 plants, functioning as the primary rate-limiting enzyme in the C4 pathway [42]. In terms of its contribution to gluconeogenesis, cytosolic PPDK uses pyruvate generated by amino acid catabolism during Arabidopsis seed germination as a substrate for gluconeogenesis [16]. Additionally, PPDK affects leaf senescence, converting pyruvate derived from amino acid catabolism to PEP, which is converted to glutamine for amino acid transport. In Arabidopsis and tobacco, the overexpression of *PPDK* during the aging process increases the seed size and seed nitrogen content. Arabidopsis plants overexpressing *At4g15530(PPDK)* during the aging process are reportedly larger than control plants [43]. PPDK was revealed to influence the fruit ripening process, as well as leaf maturation and senescence, in tomato but not in peach or pepper. This suggests that PPDK plays a role in the gluconeogenesis pathway, but this role may vary across species [44]. Mutating *LOC4338750(OsPPDKB*/*FLO4)* in *Oryza sativa* (rice) cells decreases the starch content of mutant grains, which is accompanied by a decrease in the amylose:amylopectin ratio and an increase in the lipid content [45]. These findings imply that cytosolic PPDK is an important regulator of the formation of carbon skeletons [46], including during the biosynthesis of starch and fatty acids. It is conceivable that increased PPi levels during the conversion of PEP to pyruvate by PK and PPDK may stimulate the sucrose synthase-dependent starch biosynthesis pathway [47].

The C4 pathway uses PPDK in leaf chloroplasts to regenerate the initial CO_2_ acceptor (i.e., PEP). A bifunctional regulatory protein (PDRP) controls PPDK activity. The reversible phosphorylation of the Thr residue of PPDK inactivates the enzyme. PDRP can promote the dark-induced phosphorylation and light-induced dephosphorylation of this residue, thereby controlling PPDK activation/inactivation [48,49]. Recent studies have further clarified the specific effects of PDRP on C4 crop photosynthesis. For example, the Fv/Fm value of the *PDRP* knockout *Setaria viridis* mutant gradually decreases over several days under fluctuating light conditions (i.e., repeated exposure to 1 min light phases during a 16 h photoperiod). This may be attributed to the fact that PPDK activity is not regulated, resulting in an unstable and limited supply of primary carboxylated PEP. This impairs the ability of photosynthetic organs to use excitation energy for photochemical reactions. One of the functions of PDRP in C4 photosynthesis is to maintain optimal photoreaction efficiency during changes in incident light [50].

Environmental conditions also substantially affect PPDK activity. In response to drought stress, the specific activity of tobacco PPDK increases significantly after 11 days, with increases in PPDK contents (2.7 times) also enhancing PPDK activity. The reaction catalyzed by PPDK produces ATP and Pi, resulting in a significant increase in biological energy under stress conditions. At this time, the ATP produced by mitochondria through oxidative phosphorylation may be limited, or the demand for ATP for biosynthesis reactions may increase [51].

## 2. PEP Is Involved in Plant Catabolic Activities and the Regulation of Plant Growth and Development

### 2.1. Effects of PK-Mediated PEP Degradation on the Regulation of Plant Growth and Development

Pyruvate kinase (PK) plays a significant role in PEP metabolism, as well as in the final step of the glycolytic pathway in plants. This enzyme catalyzes the transfer of a phosphate group from PEP to adenosine diphosphate, producing one molecule of pyruvate and one molecule of ATP. This reaction represents the fundamental metabolic process underlying cytosolic glycolysis and gluconeogenesis in plant cells, providing the energy necessary for sucrose synthesis, translocation, and distribution [52].

In plants, PK is divided into two distinct forms: plastidial pyruvate kinase (PKp) and cytosolic pyruvate kinase (PKc). These two isoenzymes, which are ubiquitous in plants, differ significantly in terms of physical and kinetic/regulatory properties [53], which affects crop maturation. A study of rice PKc revealed that a T-DNA insertion mutation to *OsPK1* causes dwarfism and panicle enclosure while also decreasing the seed setting rate. The sucrose content in the panicles of the *ospk1* mutant decreases by approximately 84%. These findings reflect the changes in glucose and fructose metabolism and sugar transport in the *ospk1* mutant [54]. OsPK2, which is encoded by a gene in the rice chloroplast genome, is a PKp that plays a role in the synthesis of starch in the endosperm, the formation of starch granules, and the grain filling process. The *ospk2* mutant is characterized by significant decreases in grain weight and starch content as well as altered starch physicochemical properties (relative to WT grains). The normal starch compound granules were drastically reduced and more single granules filled the endosperm cells of *ospk2*. Additionally, after a 1-year storage period, the *ospk2* seed germination rate is much lower than the WT seed germination rate [55]. OsPK3 in the rice cytoplasm is responsible for regulating the transport of sugars from the source to the sink, which in turn controls the grain filling process. A T-DNA mutant OsPK3 delays amyloplast development in endosperm cells. Moreover, the abundance of storage compounds (predominantly starch, protein, and lipids) is significantly lower in *ospk3* seeds than in WT seeds [56]. Furthermore, ABA plays a pivotal role in regulating the expression of PK-encoding genes *LOC4349774(OsPK1)*, *LOC4351493(OsPK4)*, *LOC4349454*(*OsPK9)*, and *LOC4350078(OsPK10*) in rice. In a study of PK-related genes in *Fragaria ananassa* (strawberry), the overexpression of *FxaC_15g00080* (*FaPKc2.2)* was observed to negatively regulate strawberry seed maturation and inhibit the accumulation of organic acids [57]. Earlier research on GhPKc, which is a PKc in *Gossypium hirsutum* (cotton), showed that the rapid elongation of fiber cells is accompanied by a continuous decrease in GhPKc activity. Inhibiting *KX369413(GhPK6*) expression promotes cotton fiber elongation. One potential explanation is that during the rapid elongation of cotton fibers, GhPKc activity is relatively low, which ensures that PEP is effectively used by PEPC to synthesize malic acid, thereby increasing cell turgor pressure and facilitating cell elongation [58,59,60]. Therefore, PK functions are critical for crop production.

In Arabidopsis, the energy status affects PK activities. An examination of five PK isoenzymes in Arabidopsis revealed that PKc1, PKc2, and PKc3 activities are negatively regulated by ATP. Hence, these isoenzymes are inhibited when the energy supply in vivo is sufficient. In contrast, PKc4 and PKc5 activities are unaffected by ATP, indicating that the functions of these two isoenzymes are not associated with the energy state of the cell [60]. Studies on PKc from *Ricinus communis* demonstrated that glutamate is the most effective inhibitor, whereas aspartate acts as an activator [61]. Furthermore, the TCA cycle intermediates citric acid, 2-oxoglutaric acid, fumaric acid, and malic acid can decrease the activity of certain plant PKs [60], implying that the production of pyruvate in a process involving PEP is regulated by a range of metabolic intermediates.

Environmental stress significantly influences PK activities. A *Glycine max* PK-encoding gene involved in the response to salt stress has been identified. The ectopic expression of *SoyZH13_19G000600.m1(GmPK21)* adversely affects the tolerance of Arabidopsis plants to salinity, indicating that GmPK21 may be a negative regulator of plant salt tolerance [62]. In mulberry trees, the expression levels of genes encoding PKs are significantly upregulated in response to submergence stress. This may be attributed to the fact that flooding and submergence stress directly limit oxygen exchange, thereby causing energy shortages in plants and promoting glycolysis in plants [63].

### 2.2. PEP Degradation Mediated by PEPC Regulates Plant Growth and Development

PEPC is a cytoplasmic enzyme that is ubiquitous in higher plants and also widely distributed in bacteria, cyanobacteria, and green algae [64,65]. It catalyzes the irreversible β-carboxylation of PEP in the presence of HCO_3_^−^ and Mg_2_^+^ to produce OAA and Pi. Thus, it is closely related to C4-dicarboxylic acid metabolism in plants [64]. In addition to its essential role in the initial fixation of atmospheric CO_2_ during C4 photosynthesis and C4 acid metabolism (CAM), PEPC also plays a compensatory role in various non-photosynthetic processes, including C/N allocation in C3 leaves, seed formation and germination, and fruit ripening [64]. Additionally, non-photosynthetic isoforms of PEPC play a specific role in carbon metabolism in guard cells during the opening of stomata, as well as in the formation of C4 acids in plant host cells in the root nodules of nitrogen-fixing legumes [66,67]. The disparate functions of these enzymes in photosynthetic and non-photosynthetic tissues reflect their distinct contributions to PEP catabolism.

Plant *PEPC* genes are included in a small gene family that encodes several plant-type PEPC (PTPC) and distantly related bacterial-type PEPC(BTPC) [68,69]. A class 1 PEPC in developing *R. communis* seeds (COS) was revealed to be a 410 kDa homotetramer composed of identical p107 subunits that are phosphorylated in vivo [70]. Class 1 PEPC activities in diverse plant tissues under varying physiological conditions are regulated by various mechanisms (e.g., phosphorylation, monoubiquitination, alterations in intracellular pH, and allosteric effectors). In *R. communis* plants, monoubiquitination is more prevalent than phosphorylation and appears to be the dominant post-translational modification of class 1 PEPCs [69,71]. In contrast, a class 2 PEPC in developing COS was identified as a novel 910 kDa heterooctameric complex, in which the PTPC subunit, which is identical to a subunit in the class 1 PEPC homolog, is tightly associated with four BTPC subunits. Class 2 PEPC may function in a metabolic overflow mechanism, capable of maintaining a substantial flux from PEP to L-malate under physiological conditions, thereby suppressing class 1 PEPCs [69]. It has also been postulated that a COS type 2 PEPC is capable of supporting a substantial flux of PEP to malate, which is essential for fatty acid synthesis and dominates the metabolism of developing COS white bodies. The PTPC and BTPC subunits of the type 2 PEPC complex have PEPC activities during COS development [69]. However, there is a notable difference in their respective affinities and sensitivities. Specifically, the PTPC subunit has a higher affinity for PEP and is more sensitive than the BTPC subunit, which is relatively insensitive to PTPC inhibitors, including L-malate and L-aspartate [71,72].

### 2.3. PEP Degradation Mediated by DHAPS Regulate Plant Growth and Development

DHAPS is a crucial enzyme for the biosynthesis of secondary metabolites in plants. The shikimate pathway, which comprises seven steps, is a metabolic pathway in bacteria, archaea, fungi, algae, some protozoa, and plants for the biosynthesis of folic acid and aromatic amino acids (tryptophan, phenylalanine, and tyrosine). In addition, this pathway serves as a conduit between carbohydrate metabolism and the biosynthesis of aromatic compounds [73]. In the shikimate pathway, PEP and 4-phospho-erythrose (E4P) are involved in a condensation reaction catalyzed by DHAPS that produces 3-deoxy-D-arabino-heptulosonate-7-phosphate (DAHP) and inorganic phosphate. Accordingly, DHAPS in the shikimate pathway influences PEP flux [73].

At the transcriptional level, certain transcription factors can regulate the abundance of metabolites associated with the shikimate pathway by modulating the expression of genes involved in this pathway. The *Solyc02g088190(SlMIXTA*-*like)* gene, which is expressed in the hairy roots of *Solanum lycopersicum*, encodes a protein that binds directly to the promoter region of the gene encoding 3-deoxy-7-phosphoheptulonate synthase and activates expression [74]. Induced *LOC544153 (SlDAHPS)* expression enhances the activity of the shikimate pathway and leads to the production of substrates for downstream secondary metabolism [74,75,76]. The overexpression of *PgMyb308*-like, which encodes a transcription factor in *Punica granatum*, in pomegranate hairy roots increases the accumulation of shikimic acid, aromatic amino acids, isoferulic acid, and lignin while decreasing the contents of gallic acid and its downstream product HT as well as various flavonoids [77]. These changes are attributed to an increase in the expression of *LOC116188438 (PgSDH1)*, which encodes shikimate dehydrogenase 1, an SDH isoenzyme involved in shikimate biosynthesis, and a decrease in the expression of *OWM85405.1(PgSDH4)*, an SDH isoenzyme presumed to be responsible for the production of gallic acid [78].

Arabidopsis contains three DHAPS isoenzymes (DHAPS1, DHAPS2, and DHAPS3), among which DHAPS2 is influenced by the Tyr concentration. A Tyr treatment decreases DHAPS2 contents in chloroplasts. DHAPS2 interacts with 14-3-3 proteins in the cytoplasm. This interaction, which is enhanced by a Tyr treatment, may result in the retention of DHAPS2 in the cytoplasm, thereby preventing it from functioning in chloroplasts with elevated Tyr levels [79].

In the plant developmental cycle, *DHAPS1* is predominantly expressed in mature leaves, whereas *DHAPS2* is primarily expressed in seedlings [80]. In contrast to DHAPS1 and 3, DHAPS2 is moderately active even in the absence of a reducing agent. However, it is inhibited by aromatic amino acids [80]. Considering *DHAPS2* expression is specifically upregulated in seedlings, and *DHAPS2* is co-expressed with genes involved in plastid development rather than aromatic amino acid metabolism [80], redox-independent DHAPS2 activity may be essential for maintaining the activity of the shikimate pathway in fast-growing non-photosynthetic tissues, wherein basal levels of aromatic amino acids must be maintained to support protein synthesis for growth, even in the absence of photosynthesis.

## 3. PEP Transport and Distribution Affect the Regulation of Plant Growth and Development

The orderly progression of cellular metabolism is achieved through the division of metabolic pathways between organelles. However, this compartmentalization necessitates the transport of specific key metabolites across relatively impermeable membranes. The plastid inner membrane serves as the primary site for the transport of metabolic intermediates between the plastid matrix and the cytosol, which is facilitated by specific transport proteins. PEP is imported into plastids via PPT for the production of aromatic amino acids [81,82,83,84]. Arabidopsis *At5g33320(CUE1)* encodes a PPT localized in the plastid inner membrane [81]. Notable characteristics of the homozygous *cue1*/heterozygous *eno1* mutant [*cue1*/*eno1* (+/−)] include delayed vegetative growth, disrupted floral development, and considerable seed abortion (up to 80%). Other phenotypes, such as decreased oil contents in seeds, decreased flavonoid and aromatic amino acid contents in flowers, impaired lignin biosynthesis in stems, and abnormal outer wall formation in pollen, have also been observed [85]. This suggests that PEP plays a significant role in the plastid. Compared with C3 plants, C4 plants typically have a higher photosynthetic CO_2_ uptake rate in their leaves. Consequently, C4 photosynthesis necessitates a greater capacity for transporting metabolites. According to a phylogenetic analysis of the two PPT homologs, PPT1 and PPT2, PPT1 was recruited for C4 photosynthesis in multiple C4 lineages. C4 evolution was accompanied by increases in *PPT1* transcription. Furthermore, *At4g23660(PPT1)* expression was transferred from the roots to the leaves and from bundle sheath cells to mesophyll cells. Additionally, the rapid and sustained response of *PPT1* to light has been demonstrated. One potential explanation for the elevated transcription of C4 *PPT1* is the presence of a MEM1 B submodule in its promoter region [86].

The redistribution of PEP may serve as an indicator of the energy state of plant root nodules. A restricted supply of sucrose causes root nodule cells to enter a relatively low-energy state, which is accompanied by increases in adenosine monophosphate (AMP) levels. This process promotes the formation of a GmNAS1–GmNAP1 heterodimer and leads to the accumulation of GmNFYC10 in the nucleus, thereby driving glycolysis and pyruvate production. With an additional supply of sucrose when the energy state of the root tuber increases as the AMP level decreases, GmNAS1 and GmNAP1 mainly form homodimers that maintain GmNFYC10 in mitochondria, thereby preventing it from accumulating in the nucleus. This results in decreased pyruvate production, as well as the conversion of PEP to OAA. Relatively low GmNFYC10a concentrations in the nucleus result in decreased expression of glycolysis-related genes involved in pyruvate production, which alters PEP allocation in favor of nitrogen fixation. Accordingly, the ratio of PEP converted into pyruvate and OAA is largely dependent on nodule energy state in the wild type (36:64 under a low root nodule energy status and 3:97 under a high root nodule energy status) [4].

The metabolic flux of PEP is critical for the formation of downstream carbon skeletons, including the methylerythritol phosphate pathway, shikimate and phenylpropanoid pathways, and anaplerotic pathway [3]. In response to ozone stress, the downstream energy metabolism and the demand for carbon skeletons increase because of increased metabolic activities. Therefore, the source and fate of PEP must be considered. The PEP metabolic pathway in leaves is influenced by ozone stress. An imbalance between PEP production and use may disrupt homeostasis and ultimately lead to cell death [3]. Furthermore, light influences the metabolic flux of PEP. A metabolic analysis of the rapid response of Arabidopsis to light stress revealed a rapid depletion of the PEP pool in cells. In terms of direct energy production, the conversion of PEP to pyruvate produces ATP, From the standpoint of immediate energy production, the conversion of PEP to pyruvate generates ATP that could be used as an energy source to maintain the cell at a high redox state. Therefore, PEP may play an important role in plants adapting to rapid changes in light conditions [87].

## 4. Conclusions

As shown in Table 1**,** the synthesis, metabolism, and transportation related to PEP have significant impacts on the growth and development of plants. PEP is a molecule with high-energy phosphate bonds as an energy donor and can also serve as a signaling molecule to participate in the regulation of growth and development. In this article, we have elaborated on the role of PEP in plant growth and development in terms of synthesis and degradation energy allocation, as well as signaling molecules. We have not established a comprehensive and systematic regulatory network centered on PEP to elucidate the important roles of plant cell tissue organ development, from vegetative growth to reproductive growth, from abiotic stress to biotic stress, and other processes. Our review can provide a reference for the basic and applied research of PEP in plants in the future, and we look forward to more research focusing on the important role of PEP in plant growth and development in the future.

## Figures and Tables

**Figure 1 ijms-26-00391-f001:**
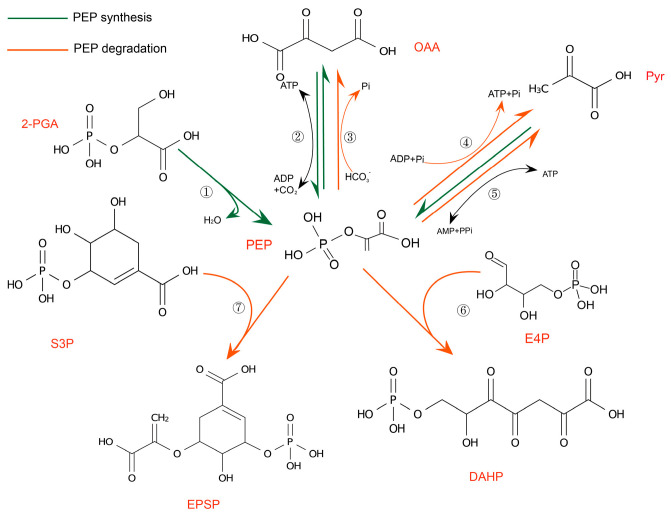
PEP synthesis and metabolism pathways in plants [6]: ➀ ENO (EC 4.2.1.11), ➁ PEPCK (EC 4.1.1.49), ➂ PK (EC 2.7.1.40), ➃ PEP (EC 4.1.1.31), ➄ PPDK (EC 2.7.9.1), ➅ DAHPS (EC 2.5.1.54), ➆ 3-phosphoshikimate 1-carboxyviniltransferase (EC 2.5.1.19).

**Table 1 ijms-26-00391-t001:** Phenotypic of PEP-related metabolism and transport processes.

PEP-Related Metabolism and Transport Processes	Gene	Mutation Type/Stress Conditions	Common Name	Latin Name	Phenotypic	Year	Reference
ENO	*AtENO1*	T-DNA insertion	Thale cress	*Arabidopsis thaliana*	distorted trichomes and fewer root hairs	2009	[7]
ENO	*AtENO2*	T-DNA insertion	Thale cress	*Arabidopsis thaliana*	small leaves, short siliques, increased sensitivity to abiotic stresses	2021	[9]
PEPCK	*AtPEPCK*	antisense *35S-PEPCK1*	Thale cress	*Arabidopsis thaliana*	soluble sugar content, stunted seedling growth, cotyledons are unable to unfold, roots cannot elongate	2004	[17]
PEPCK	*SlPEPCK*	RNA interference	tomato	*Solanum lycopersicum*	decrease in the sugar content and accumulation of malate	2015	[19]
PEPCK	*CaPEPCK*	CaPEPCK-overexpressing	chilli	*Capsicum annuum*	grow faster and produce larger leaves	2015	[20]
PEPCK	*ZmPEPCK*	sufficient light	maize	*Zea mays*	ZmPEPCK phosphorylation increases as the light intensity increases	2023	[25]
PEPCK	*AtMC9*	MC9 overexpressing	Thale cress	*Arabidopsis thaliana*	hypocotyls grow longer and enhanced PEPCK activity	2013	[30]
PPDK	*PDRP*	PDRP knockout, fluctuating light conditions	green foxtail	*Setaria viridis*	the Fv/Fm gradually decreases over several days	2022	[50]
PPDK	*OsPPDKB*	single base conversion	rice	*Oryza sativa*	decrease in the amylose:amylopectin ratio and an increase in the lipid content	2022	[45]
PK	*OsPK2*	OsPK2 loss of function	rice	*Oryza sativa*	significant decreases in grain weight and starch	2018	[55]
PK	*OsPK3*	T-DNA insertion	rice	*Oryza sativa*	delays amyloplast development in endosperm cells; abundance of storage compounds (predominantly starch, protein, and lipids) is significantly lower in seeds	2020	[56]
PK	*FaPKc2.2*	FaPKc2.2 overexpression	strawberry	*Fragaria × ananassa*	negatively regulate strawberry seed maturation and inhibit the accumulation of organic acids	2022	[57]
PK	*GhPK6*	GhPK6 silenced	cotton	*Gossypium hirsutum*	increased fiber cell elongation and reduced reactive oxygen species (ROS) accumulation	2010	[59]
PK	*GmPK21*	Gene ectopic expression in Arabidopsis thaliana	Thale cress	*Arabidopsis thaliana*	decreased in tolerance of Arabidopsis plants to salinity	2024	[62]
DHAP	*SlMIXTA-like*	*SlMIXTA*-like overexpressing	tomato	*Solanum lycopersicum*	activates SlDHAPS expression	2020	[74]
DHAP	*PgMyb308-like*	PgMyb308-like overexpression	pomegranate	*Punica granatum*	increases the accumulation of shikimic acid, aromatic amino acids, isoferulic acid, and lignin, decreasing the contents of gallic acid and its downstream product HT, as well as various flavonoids	2022	[77]
PEP transport and distribution	*CUE1*	CUE1 knockout	Thale cress	*Arabidopsis thaliana*	delayed vegetative growth, disrupted floral development, considerable seed abortion decreased oil contents in seeds, decreased flavonoid and aromatic amino acid contents in flowers, impaired lignin biosynthesis in stems, and abnormal outer wall formation in pollen	2010	[85]
PEP transport and distribution	*GmNAS1, GmNAP1*	Sucrose,*GmNAS1 GmNAP1* double knockout	Soybean	*Glycine Soja*	2-phosphoglycerate (2-PG) and phosphoenolpyruvate (PEP) contents pyruvate lower and oxaloacetic acid (OAA) contents higher	2022	[4]

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
