# Peer review of "Phosphoenolpyruvate and Related Metabolic Pathways Contribute to the Regulation of Plant Growth and Development"

_ijms, 2025, doi:10.3390/ijms26010391_

Round 1

Reviewer 1 Report

Comments and Suggestions for Authors

Dear Editor,

I have reviewed the manuscript titled “Phosphoenolpyruvate and related metabolites contribute to the regulation of plant growth and development” by Hu et al. While the topic is highly relevant to plant research, the current version of the manuscript does not meet the standards required for publication. Below, I detail the reasons for my recommendation to reject the manuscript in its present form.

Major Concerns

  1. Figure 1:
    • This figure is incomplete, as it does not include all enzymes involved in PEP metabolism. Additionally, the image closely resembles Figure 1 from the review by Rojas and Iglesias (2023, AoB PLANTS 15:1–11), which the authors have not cited. The PEPCK section also shows significant similarities to the aforementioned review, raising concerns about originality.
    • The depiction of reactions lacks clarity regarding their reversibility or irreversibility. For instance, the phosphoenolpyruvate carboxylase (PEPC) reaction is irreversible, whereas the phosphoenolpyruvate carboxykinase (PEPCK) reaction is reversible. These details are essential for understanding PEP metabolism.
    • Dividing the reactions into PEP "metabolism" and "synthesis" is misleading, as PEP metabolism is highly flexible and context-dependent, adapting to various physiological conditions.
  2. Gene Nomenclature:
    • Throughout the text, when referring to genes, the authors should include locus identifiers to avoid ambiguity.
  3. Line 35:
    • The reference to PEP’s role in animal diseases is irrelevant to the scope of this review, which focuses on plant metabolism.
  4. Line 95-96:
    • The statement regarding plant PEPCKs being GTP-dependent is incorrect. Plant PEPCKs are ATP-dependent.
  5. Line 135:
    • The claim that "PEPCK can regulate its own activity" is incorrect. Research has shown that the cysteine protease MC9 is responsible for cleaving PEPCK’s N-terminal, regulating its activity.
  6. Figure 2:
    • This figure provides little useful information. A table format would be more effective, listing the involved genes, their phenotypic/metabolic effects, and corresponding references.
  7. Lines 100-104:
    • The assertion that PEPCK mutants cannot develop into mature plants is inaccurate. While these mutants exhibit growth defects, they are capable of reaching maturity. Additionally, the manuscript omits the role of pyruvate, phosphate dikinase (PPDK) in bypass pathways and fails to explain the work of Eastmond et al., who identified a novel gluconeogenic route in Arabidopsis.
  8. Line 114-115:
    • The classification of 3-phosphoglycerate (3-PGA) as a downstream product of glycolysis is incorrect. In photosynthetic tissues, it is a direct product of the Calvin-Benson-Bassham cycle, and in non-photosynthetic tissues, it is located at the middle of glycolysis. Also it would be informative to explain the suggested role of 3-PGA on plant inmmunity.
  9. Line 384:
    • The phrase "downstream carbon skeletons" is vague and lacks precision. A more specific explanation is needed.

Minor Concerns and Grammatical Errors

  1. Lines 69-70: Replace "This series of reactions" with "These reactions" for clarity.
  2. Line 69: Add a period after [10].
  3. Line 76: Adjust the spacing between parentheses and "NLRs."
  4. Line 81: Rephrase to “... shorter siliques, less chlorophyll, and flowers earlier…”.
  5. Line 94: Use the correct sequence: "fungi, yeast, and bacteria."
  6. Line 239: Revise to “The T-DNA mutant OsPK3.”

Recommendation

The manuscript addresses an important area of plant metabolism research, but it requires significant revisions to improve its scientific accuracy, clarity, and presentation. I encourage the authors to address the issues outlined above, particularly the need for accurate references, precise terminology, and clearer figures. I recommend that the manuscript be rejected in its current form but suggest resubmission after thorough revisions.

Author Response

Reviewer #1: I have reviewed the manuscript titled “Phosphoenolpyruvate and related metabolites contribute to the regulation of plant growth and development” by Hu et al. While the topic is highly relevant to plant research, the current version of the manuscript does not meet the standards required for publication. Below, I detail the reasons for my recommendation to reject the manuscript in its present form.

Response: Thank you for your professional and insightful advice. In fact, we have learned a lot from your comments and further revised our paper based on your feedback. We hope our revisions will satisfy you. Thank you again for your professional guidance.

Line 24 Figure 1:

This figure is incomplete, as it does not include all enzymes involved in PEP metabolism. Additionally, the image closely resembles Figure 1 from the review by Rojas and Iglesias (2023, AoB PLANTS 15:1–11), which the authors have not cited. The PEPCK section also shows significant similarities to the aforementioned review, raising concerns about originality.

The depiction of reactions lacks clarity regarding their reversibility or irreversibility. For instance, the phosphoenolpyruvate carboxylase (PEPC) reaction is irreversible, whereas the phosphoenolpyruvate carboxykinase (PEPCK) reaction is reversible. These details are essential for understanding PEP metabolism.

Dividing the reactions into PEP "metabolism" and "synthesis" is misleading, as PEP metabolism is highly flexible and context-dependent, adapting to various physiological conditions.

Response: Corrected. References have been added, and reversibility of reactions has been added. Existing studies all exist in archaea, and there is no evidence that PEP synthase exists in plants. We also believe that the synthesis and decomposition of PEP are inseparable, but there is currently no better perspective to separately explain the roles of these functions in plants. Therefore, we have retained the chapters on synthesis and decomposition.

Gene Nomenclature. Throughout the text, when referring to genes, the authors should include locus identifiers to avoid ambiguity.

Response: Corrected, Gene identifier added

Line 35:The reference to PEP’s role in animal diseases is irrelevant to the scope of this review, which focuses on plant metabolism.

Response: Corrected. The immunity part has already deleted

Line 95-96: The statement regarding plant PEPCKs being GTP-dependent is incorrect. Plant PEPCKs are ATP-dependent.

Response: Corrected

Line 135: The claim that "PEPCK can regulate its own activity" is incorrect. Research has shown that the cysteine protease MC9 is responsible for cleaving PEPCK’s N-terminal, regulating its activity.

Response: Corrected.

Figure 2: This figure provides little useful information. A table format would be more effective, listing the involved genes, their phenotypic/metabolic effects, and corresponding references.

Response: Thanks for your suggestion. We have change are picture into the table format

Lines 100-104: The assertion that PEPCK mutants cannot develop into mature plants is inaccurate. While these mutants exhibit growth defects, they are capable of reaching maturity. Additionally, the manuscript omits the role of pyruvate, phosphate dikinase (PPDK) in bypass pathways and fails to explain the work of Eastmond et al., who identified a novel gluconeogenic route in Arabidopsis.

Response: Thanks for your suggestion. The incorrect statement has been revised and the new gluconeogenesis pathway discovered by Eastmond has been added.

Line 114-115:The classification of 3-phosphoglycerate (3-PGA) as a downstream product of glycolysis is incorrect. In photosynthetic tissues, it is a direct product of the Calvin-Benson-Bassham cycle, and in non-photosynthetic tissues, it is located at the middle of glycolysis. Also it would be informative to explain the suggested role of 3-PGA on plant inmmunity.

Response: Thanks for your suggestion. Downstream products have been changed to intermediate products. We have added the role of triphosphoglycerate in plant immunity

Line 384: The phrase "downstream carbon skeletons" is vague and lacks precision. A more specific explanation is needed.

Response: Thanks for your suggestion. We have explain the downstream carbon skeleton

Minor Concerns and Grammatical Errors

Lines 69-70: Replace "This series of reactions" with "These reactions" for clarity.

Response:correceted

Line 69: Add a period after [10].

Response:correceted

Line 76: Adjust the spacing between parentheses and "NLRs."

Response: correceted

Line 81: Rephrase to ... shorter siliques, less chlorophyll, and flowers earlier…”.

Response: correceted

Line 94: Use the correct sequence: "fungi, yeast, and bacteria."

Response: correceted

Line 239: Revise to The T-DNA mutant OsPK3.

Response: correceted

Reviewer 2 Report

Comments and Suggestions for Authors

Thank you for sending me this manuscript by Runzhou Hu, et al.  The authors provide a comprehensive summary of the metabolic activities of PEP in plants, including embryo formation, root development, synthesis of secondary metabolites, and the formation of lignification. Authors also summarize new findings including PEP's role in nodule energy sensing and carbon allocation under the influence of ozone.

There are issues that are listed in order, as follows:

1) in the abstract section, there are extra words (background, result). In addition, the abstract should be improved, since it is very short.

 2) page 1, line 36-27: author states that PEP contains the highest-energy phosphate bond in living organisms. Please provide references for this statement.

3) page 1, line 43-44: it seems that the word figure 1 is not in the correct position. Please fix this error.

4) page 2, line 49: It seems the words main text should be removed.

5) page 2, line 69: please add a dot after the reference [10].

6) the conclusion is very short; it would be improved by adding more details including future directions in the field of ENO studies.

7) Author described function, regulation and pathways where ENO is involved. But it would be interesting to see a figure of ENO crystal or cryoEM structures to make this manuscript more complete.

Author Response

Reviewer #2: Thank you for sending me this manuscript by Runzhou Hu, et al.  The authors provide a comprehensive summary of the metabolic activities of PEP in plants, including embryo formation, root development, synthesis of secondary metabolites, and the formation of lignification. Authors also summarize new findings including PEP's role in nodule energy sensing and carbon allocation under the influence of ozone.

Response: Thank you for your suggestions and guidance. We have made further revisions to the manuscript according to your requirements and hope you are satisfied.

In the abstract section, there are extra words (background, result). In addition, the abstract should be improved, since it is very short.

Response: We have refined the abstract as required

Line 36-37: author states that PEP contains the highest-energy phosphate bond in living organisms. Please provide references for this statement.

Response: Thanks for your suggestion. The reference has Provieded

Line 43-44: it seems that the word figure 1 is not in the correct position. Please fix this error.

Response: correceted

Line 49: It seems the words main text should be removed.

Response: Corrected

Line 69: please add a dot after the reference [10].

Response: Corrected

he conclusion is very short; it would be improved by adding more details including future directions in the field of ENO studies.

Response: Thanks for your suggestion, We have provided a more detailed explanation.

Author described function, regulation and pathways where ENO is involved. But it would be interesting to see a figure of ENO crystal or cryoEM structures to make this manuscript more complete.

Response: Thanks for your suggestion. According to our search, including in databases such as uniprot, we have not yet found the crystal structure of ENO1. In fact, our laboratory has also discovered a series of ENO proteins in rice. We will investigate the crystal structures of these ENO proteins in future research. Please look forward to our further studies.

Round 2

Reviewer 1 Report

Comments and Suggestions for Authors

Dear Editor,

I have reviewed the revised manuscript titled “Phosphoenolpyruvate and related metabolites contribute to the regulation of plant growth and development” by Hu et al. While the authors have addressed many of my previous comments, the current version of the manuscript does not meet the standards required for publication. The main issues include inconsistencies in the text, excessive use of non-scientific terminology, and imprecise definitions of key terms. Below, I detail the reasons for my recommendation to reject the manuscript in its present form.

Major Issues

Title:
In the title, the authors claim to review “Phosphoenolpyruvate and related metabolites.” This is a vague statement, as the text does not clearly define what these related metabolites are. Please either provide a definition or revise the manuscript title to better reflect the content.

Figure 1 and overall text:
In biochemistry, metabolic enzymatic reactions are typically classified as anabolic vs. catabolic or as synthesis vs. degradation. The use of the term “decomposition” is inconsistent with accepted scientific terminology and is confusing. For instance, the conversion of PEP into other metabolites can occur as part of an anabolic pathway, which does not align with the concept of “decomposition.” Please revise this terminology throughout the manuscript.

Line 51-53:
In the title of section 1.1, ENO is described as a “decomposition enzyme,” whereas in the first line of the subsequent paragraph it is referred to as a “PEP synthesis enzyme.” These statements are contradictory and need clarification.

Line 80:
Define “2/3-PG” to avoid confusion. Additionally, avoid introducing abbreviations that are used only once in the text. This recommendation applies to the rest of the manuscript as well.

Line 92:
The description of PEPCK as a “PEP synthesis enzyme” is incorrect. It has been clearly demonstrated that this enzyme is reversible and plays a versatile role under different metabolic contexts, such as glycolysis and gluconeogenesis. Please revise this statement for accuracy.

Lines 105-106:
The sentence is unclear and does not use scientific vocabulary. Please rephrase for clarity and precision.

Line 121:
The reference to a “symbiotic relationship between phytoplankton and bacteria” and its connection to plant immunity is unclear. Please elaborate on this point with more detail.

Line 208:
What does “aging process” mean in the context of plants? Does it refer to senescence? Please clarify.

Lines 255-256:
What does “monosaccharide metabolism” include? Does it refer only to glucose and fructose? Please specify.

Lines 315-316:
Define and describe what PTPCs and BTPCs are. As previously mentioned, avoid single-use abbreviations.

Line 421:
Define the MEP pathway. Again, avoid single-use abbreviations.

Line 430-431:
The authors hypothesize that substrate-level phosphorylation during the conversion of PEP to pyruvate is sufficient to maintain the cell in a highly reduced state. Please elaborate on this hypothesis and provide supporting evidence or references.

Line 451:
Remove the statement “our review is still not comprehensive enough.” Instead, include open questions and future research directions to strengthen the conclusion.

Minor Issues

Line 12:
Change “play” to “plays.”

Line 18:
Change “further displayed” to “display.”

Line 19:
Change “provided” to “and provides,” and “PEP” to “PEP metabolism.”

Line 79:
Change “affects” to “affect.”

Line 422:
Add “and” before “anaplerotic.”

Line 445:
Add a comma before “and.”

Line 446:
Add “a” before “with.”

Line 448:
Remove the sentence “its importance is increasingly being recognized by researchers,” as it lacks scientific tone and is not supported by referenced data.

I consider that by addressing the issues outlined above, the authors can significantly improve the clarity and scientific rigor of their manuscript.

Author Response

Dear Editor,

I have reviewed the revised manuscript titled “Phosphoenolpyruvate and related metabolites contribute to the regulation of plant growth and development” by Hu et al. While the authors have addressed many of my previous comments, the current version of the manuscript does not meet the standards required for publication. The main issues include inconsistencies in the text, excessive use of non-scientific terminology, and imprecise definitions of key terms. Below, I detail the reasons for my recommendation to reject the manuscript in its present form.

Response: Thank you again for your detailed and professional advice. We have made revisions to the manuscript based on your feedback regarding the existing issues.

Major Issues

Title:

In the title, the authors claim to review “Phosphoenolpyruvate and related metabolites.” This is a vague statement, as the text does not clearly define what these related metabolites are. Please either provide a definition or revise the manuscript title to better reflect the content.

Response: Corrected. Your comments are very useful, we are focusing on metabolic pathways related to PEP, so we have changed ‘metabolites’ to ‘metabolic pathways’.

Figure 1 and overall text:

In biochemistry, metabolic enzymatic reactions are typically classified as anabolic vs. catabolic or as synthesis vs. degradation. The use of the term “decomposition” is inconsistent with accepted scientific terminology and is confusing. For instance, the conversion of PEP into other metabolites can occur as part of an anabolic pathway, which does not align with the concept of “decomposition.” Please revise this terminology throughout the manuscript.

Response: Corrected, we have modified it to PEP synthesis and degradation.

Line 51-53:

In the title of section 1.1, ENO is described as a “decomposition enzyme,” whereas in the first line of the subsequent paragraph it is referred to as a “PEP synthesis enzyme.” These statements are contradictory and need clarification.

Response: Corrected.

Line 80:

Define “2/3-PG” to avoid confusion. Additionally, avoid introducing abbreviations that are used only once in the text. This recommendation applies to the rest of the manuscript as well.

Response: Corrected.

Line 92:

The description of PEPCK as a “PEP synthesis enzyme” is incorrect. It has been clearly demonstrated that this enzyme is reversible and plays a versatile role under different metabolic contexts, such as glycolysis and gluconeogenesis. Please revise this statement for accuracy.

Response: Corrected, Due to the Reversibility of the reaction caused by PEPCK we changed it into synthesis and degradation.

Lines 105-106:

The sentence is unclear and does not use scientific vocabulary. Please rephrase for clarity and precision.

Response: Thank you for your interest in the scientific statements in our article, we have reworked our statements to ensure that they are scientifically correct.

Line 121:

The reference to a “symbiotic relationship between phytoplankton and bacteria” and its connection to plant immunity is unclear. Please elaborate on this point in more detail.

Response: Thank you for your attention to the rigour of the literature in the article. After reviewing our initial manuscript, I found that it is glycerol-3-phosphate that plays a role in plant immunity, not 3-phosphoglycerate, we have made a correction to this error in the manuscript

Line 208:

What does “aging process” mean in the context of plants? Does it refer to senescence? Please clarify.

Response: Thank you for your suggestion. it refers to senescence, we have made this correction

Lines 255-256:

What does “monosaccharide metabolism” include? Does it refer to glucose and fructose? Please specify.

Response: Thank you for your suggestion. It refers only to glucose and fructose we have make this correction.

Lines 315-316:

Define and describe what PTPCs and BTPCs are. As previously mentioned, avoid single-use abbreviations.

Response: Corrected, we have defined PTPC and BTPC.s here refers to the plural form.

Line 421:

Define the MEP pathway. Again, avoid single-use abbreviations.

Response: Thank you for the heads up. We have defined the MEP pathway.

Line 430-431:

The authors hypothesize that substrate-level phosphorylation during the conversion of PEP to pyruvate is sufficient to maintain the cell in a highly reduced state. Please elaborate on this hypothesis and provide supporting evidence or references.

Response: Thank you for your suggestion. After reviewing the reference we found that ‘highly reduced state’ is a mistake using we have changed it into high redox state.

Line 451:

Remove the statement “our review is still not comprehensive enough.” Instead, include open questions and future research directions to strengthen the conclusion.

  Response: Thanks, Corrected. In fact, our recent study found that PEP plays an important role in regulating plant development. We found that external application or alteration of endogenous PEP synthesis levels in plants can affect plant growth and development. We are currently studying the relevant mechanisms. At the same time, we also noticed a study in Science (DOI: 10.1126/science.abq8591) that reported that energy allocation related to pep can affect the occurrence of rhizobia in plants under the condition of nodules. Therefore, in this paper, we attempt to clarify the effects of PEP on plant growth and development, providing clues for our own research and hoping that more researchers will pay attention to the role of PEP.

Round 3

Reviewer 1 Report

Comments and Suggestions for Authors

I do not have additional comments for this manuscript. I recommend the publication as the authors have attended to all my comments and suggestions.